# A ‘Spicy’ Mechanotransduction Switch: Capsaicin-Activated TRPV1 Receptor Modulates Osteosarcoma Cell Behavior and Drug Sensitivity

**DOI:** 10.3390/ijms26188816

**Published:** 2025-09-10

**Authors:** Arianna Buglione, David Becerril Rodriguez, Simone Dogali, Giulia Alloisio, Chiara Ciaccio, Marco Luce, Stefano Marini, Luisa Campagnolo, Antonio Cricenti, Magda Gioia

**Affiliations:** 1Department of Clinical Sciences and Translational Medicine, University of Rome Tor Vergata, 00133 Rome, Italy; 2Department of Sciences, Roma Tre University, 00154 Rome, Italy; 3Institute of Structure Matter del Consiglio Nazionale delle Ricerche ISM-CNR, 00133 Rome, Italy; marco.luce@artov.ism.cnr.it (M.L.);; 4Department of Biomedicine and Prevention, University of Rome Tor Vergata, 00133 Rome, Italy; campagnolo@med.uniroma2.it

**Keywords:** osteosarcoma, TRPV1, mechanotransduction, capsaicin, Src kinase, histone H3 acetylation, cell adhesion, cell reorientation, cyclic stretch, mechanosensitive ion channels

## Abstract

Osteosarcoma (OS), the most common primary malignant bone tumor, arises in highly mechanosensitive tissue and exhibits marked heterogeneity and resistance to conventional therapies. While molecular drivers have been extensively characterized, the role of mechanical stimuli in OS progression remains underexplored. Here, we identify the transient receptor potential vanilloid 1 (TRPV1) channel as a key regulator of mechanotransduction and drug responsiveness in OS cells. Using uniaxial cyclic stretch, we show that aggressive U-2 OS cells undergo TRPV1-dependent perpendicular reorientation, unlike the inert SAOS-2 cells. Confocal microscopy, immunohistochemistry, and atomic force microscopy reveal that nanomolar concentrations of capsaicin—a well-characterized TRPV1 agonist—chemically mimic this mechanical phenotype, altering metastatic traits including adhesion, edge architecture, migration, nuclear-to-cytoplasmic ratio, and sensitivity to doxorubicin and cisplatin. TRPV1 activation, whether mechanical or chemical, induces subtype-specific effects absent in healthy hFOB osteoblasts. Notably, it differentially regulates nuclear localization of the proto-oncogene Src in U-2 OS versus SAOS-2 cells. Corresponding changes in Src and acetylated histone H3 (acH3) levels support a role for TRPV1 in modulating the Src–acH3 mechanosignaling axis. These effects are tumor-specific, positioning TRPV1 as a mechanosensitive signaling hub that integrates mechanical and chemical cues to drive epigenetic remodeling and phenotypic plasticity in OS, with potential as a therapeutic target in aggressive, drug-resistant subtypes

## 1. Introduction

Cell mechanobiology is an emerging interdisciplinary field that investigates how cells perceive and respond to physical cues such as pressure, tension, stretch, and tissue stiffness. Increasing evidence suggests that mechanical alterations play a critical role in cancer initiation, progression, and metastasis [1,2]. Notably, alterations in cellular mechanotypes—reflected at the molecular level within the cytoskeleton, nucleoskeleton, and extracellular matrix—have been identified as a shared hallmark across diverse cancer types, representing a new frontier in cancer biology [3].

Understanding how mechanical forces influence cellular biochemistry and behavior—an area known as mechanocontrol—provides critical insight into cancer biology. This perspective reveals mechanisms of tumor progression that may be missed when considering only genetic or biochemical factors. The dynamic interplay between mechanical, genetic, and biochemical cues plays a decisive role in cancer development and metastasis [4]. Investigating this triad not only deepens our understanding of cancer biology but also paves the way for the development of more precise prognostic tools and innovative therapeutic strategies tailored to individual patients, thereby advancing the goals of personalized medicine.

Among the emerging areas of interest in cancer mechanobiology, mechanoreceptors—somatosensory receptors that transduce mechanical stimuli into intracellular biochemical signals via mechanically gated ion channels—were reported to play a significant role in regulating tumor cell behavior. Mechanosensitive (MS) ion channels were identified as key components of the cellular tensegrity architecture, acting as primary mechanoreceptors that converted mechanical cues into biochemical responses [5,6]. Through direct physical coupling with cytoskeletal elements, MS channels were shown to detect and respond to subtle changes in mechanical load, positioning them as essential mediators of cellular mechanotransduction [7].

At the nanoscale, mechanical forces were found to induce conformational changes in structural and regulatory cellular components—including proteins, ion channels, and chromatin—which in turn modulated protein activity, gene expression, and, ultimately, cell behavior [2,8,9]. These effects were reported to be particularly pronounced in cancer, where aberrant mechanotransduction contributed to tumor progression, metastasis, and resistance to therapy [10,11,12,13,14]. Importantly, the cellular response to mechanical stimuli was described as highly context- and cell-type-specific, influencing key processes such as proliferation, cytoskeletal organization, adhesion dynamics, signaling pathway activation, membrane mechanics, and cell volume regulation [15,16,17].

In recent years, MS ion channels have gained significant attention for their ability to translate mechanical forces from the tumor microenvironment into biochemical signals that influence cancer cell behavior. Among the most well-studied MS channels in this context were Piezo1, transient receptor potential vanilloid 1 (TRPV1), and transient receptor potential ankyrin 1 (TRPA1). These channels were described as being activated by a range of mechanical stresses—such as compression, stretch, and shear—and were shown to mediate calcium influx, thereby initiating downstream signaling cascades that regulated proliferation, migration, invasion, and therapy resistance [18,19,20,21,22,23].

Piezo1 was strongly implicated in tumor growth and metastasis through its regulation of mechanotransduction pathways and cytoskeletal remodeling [24]. Similarly, TRPV1 and TRPA1, although originally characterized for their roles in pain perception and inflammation, were increasingly recognized for their involvement in cancer cell adaptation to mechanical and oxidative stress [25,26,27]. These channels are now considered essential components that enables tumor cells to sense and respond to the dynamic mechanical landscape of solid tumors [28]. Osteosarcoma, the most common primary malignant bone tumor—has been recognized for exhibiting distinctive features that make it an excellent model for investigating how cancer cells perceive and exploit mechanical cues [29]. Importantly, OS often arises during adolescence—a developmental period marked by rapid skeletal growth and dynamic remodeling of the extracellular matrix (ECM). OS originates in bone, the most mechanically active tissue in the body, where resident cells are routinely subjected to uniaxial stretching and compressive forces due to weight-bearing and locomotion [30].

Despite therapeutic advances, OS has remained a highly aggressive tumor characterized by poor clinical outcomes, rapid disease progression, and resistance to conventional chemotherapies [29,31,32,33,34]. One hallmark of OS cells has been their altered nucleoskeletal architecture relative to healthy osteoblasts, particularly in the expression and organization of structural proteins such as lamins and emerin, which are essential for nuclear integrity and mechanotransduction [35,36].

To investigate OS cell mechanobiology, our group developed a standardized uniaxial cyclic stretch protocol (24 h, 1 Hz, 0.5% elongation) that mimics in vitro physiological mechanical forces experienced by bone cells in vivo. This mechanical stretching model has been widely adopted in vitro to study how bone-derived tumors respond and reorient themselves to cyclic mechanical loading [17,37,38,39,40,41].

Using this approach, we demonstrated that a physiologically relevant stretch regimen selectively affected osteosarcoma cells while sparing healthy human osteoblasts (hFOB) [32,42]. Specifically, cyclic stretch induced phenotypic and behavioral alterations in two OS cell lines: SAOS-2, a moderately differentiated and less aggressive line, and U-2 OS, a poorly differentiated and highly aggressive line. In contrast, hFOB cells showed negligible responses, indicating that malignant osteoblastic cells exhibit subtype-specific alterations in mechanosensitivity. These findings support the notion that OS cells reprogram normal mechanosensing pathways to support tumor progression.

In our most recent studies, we further demonstrated that mechanical activation of MS ion channels led to a non-canonical activation of Src kinase in osteosarcoma cells [42]. Src was previously reported to function as a central node in cellular mechanotransduction, acting as a key integrator of cytoskeletal tension and converting mechanical stimuli into biochemical signals that regulated cell behavior [43]. Supporting this, recent findings in macrophages identified a cytoskeleton-dependent Src–H3 acetylation axis, linking mechanical inputs to epigenetic remodeling, altered morphology, and changes in cell motility [44]. Previously, we observed that cyclic stretch-induced Src activation in OS cells correlated with increased cell migration, altered adhesion dynamics, and enhanced sensitivity to doxorubicin-induced apoptosis, while having minimal impact on proliferation. These effects underscored the functional significance of mechanosensitive Src signaling in driving malignancy-related behaviors in OS.

Moreover, the subcellular localization of Src was described as being dynamically regulated by mechanical cues [45]. Mechanical stimulation was shown to induce Src redistribution from the plasma membrane to the cytosol, and, under specific conditions, to the nucleus [46]. Such translocation events were reported to be particularly relevant in pathological contexts—including fibrosis and cancer—where nuclear Src was implicated in transcriptional regulation [36,47]. Collectively, these findings supported the existence of a tumor-specific, mechanosensitive Src signaling axis in osteosarcoma, which warrants further investigation. This pathway was not only reported to contribute to the aggressive phenotype of OS cells [42] but also represents a promising target for mechanobiology-informed therapeutic strategies [45].

Building on our previous findings—which demonstrated that LE135, a dual agonist of the mechanosensitive ion channels TRPV1 and TRPA1, could chemically replicate the cellular effects of mechanical stimulation in human osteosarcoma cells [42,48]—the present study sought to further investigate the chemical modulation of MS channel-dependent mechanosignaling in OS by including agonists for both channels, thereby clarifying their individual and combined contributions to this signaling pathway.

To investigate the role of TRPV1 and TRPA1 in OS mechanobiology, we conducted a targeted screening of mechanosensitive ion channel agonists and antagonists, assessing their capacity to mimic or inhibit the cellular responses typically induced by mechanical stimulation.

Among the TRPV1 activators, we selected capsaicin, a natural compound derived from chili peppers, based on the following: (1) its status as a well-characterized TRPV1 agonist [49,50,51,52], (2) its physiological relevance as a nutraceutical, and (3) its reported anti-cancer effects in several models, including OS [53,54,55].

In the present study, we compared the effects of MS channel activation and inhibition on U-2 OS, SAOS-2, and hFOB cells by evaluating morphological and molecular alterations using a multimodal approach. This included confocal and atomic force microscopy (AFM) and functional assays to assess changes in metastatic behaviors, such as cell adhesion, migration, and chemosensitivity of two of standard MAP chemotherapy agents—doxorubicin and cisplatin.

Crucially, we investigated whether the cytoskeleton-dependent Src–H3 acetylation axis, previously identified by our group and others as a key mediator of cellular mechanosensation, could be activated by chemical stimulation of MS channels, and whether this response converges with the signaling triggered by mechanical stimuli, such as cyclic stretch.

Together, these approaches provided a powerful framework for dissecting the roles of TRPV1 and TRPA1 in MS channel-mediated signaling. By establishing parallels between chemical and mechanical activation, this work offered novel insights into how cancer cells exploited mechanotransduction to support tumor progression, metastatic potential, and therapy resistance in mechanically dynamic microenvironments.

## 2. Results

### 2.1. TRPV1 Antagonism Counteracts Mechanically Induced Responses in Osteosarcoma Cells

Building on our previous findings that the mechanical phenotype of OS cells can be chemically mimicked using LE135, a dual agonist of TRPV1 and TRPA1 [42], we aimed to dissect the individual contributions of these MS ion channels. Specifically, we investigated whether selective inhibition of TRPV1 or TRPA1 is sufficient to replicate the broad inhibitory effects previously observed with GsMTx4, a general MS channel blocker.

To this end, OS cells were exposed to cyclic uniaxial mechanical stretch in the presence or absence of a selective-specific antagonist. An immediate-response analysis was conducted immediately following a 24 h stretch period. Cells were examined on silicone membranes using immunofluorescence and confocal microscopy to evaluate acute morphological and molecular responses to mechanical stimulation (Appendix A).

To distinguish the mechanotransductive roles of TRPV1 and TRPA1, we treated cells with 560 nM of AMG9810 (TRPV1 inhibitor) or HC030031 (TRPA1 inhibitor) during stretch exposure. Nuclear size, quantified via Hoechst staining, served as a reporter for mechanotransduction-driven morphological change. As shown in Figure 1A, AMG completely blocked the stretch-induced increase in nuclear area in U-2 OS cells (*p* < 0.0001) and reversed approximately 70% of the increase observed in SAOS-2 cells (*p* < 0.0001). In contrast, HC030031 had no significant effect on U-2 OS cells, while it can partially abrogate changes induced in SAOS-2 cells (*p* < 0.05), underscoring a predominant role for TRPV1 in mediating nuclear morphological responses to mechanical cues. Additionally, TRPV1 inhibition restored the nuclear-to-cytoplasmic (N/C) ratio—a surrogate marker for tumor aggressiveness—by approximately 70% after stretch in U-2 OS cells (*p* < 0.01) and fully reverted it to baseline levels in SAOS-2 cells (*p* < 0.05; Figure 1B). In contrast, TRPA1 inhibition had no effect in either cell model.

Next, we performed a lasting-response analysis to evaluate how TRPV1 inhibition impacts downstream functional outcomes of mechanical preconditioning. OS cells, pre-stretched with or without AMG, were subjected to a panel of assays assessing metastatic potential (Appendix A). Wound healing assays demonstrated that stretch-enhanced migratory capacity was significantly reduced by AMG by 75% in U-2 OS cells while completely reversed back to the base migration in SAOS2 cell (Figure 1C, *p* < 0.01 in both cell lines). Adhesion assays showed that TRPV1 inhibition completely reversed the stretch-induced increase in adhesion observed in U-2 OS cells (*p* < 0.0001), as well as the decrease seen in SAOS-2 cells (*p* < 0.05), indicating a cell line-specific regulatory effect (Figure 1D). From a therapeutic perspective, we previously reported that mechanical stretch modulates chemosensitivity in a cell line-specific manner: it enhances doxorubicin efficacy in SAOS-2, but induces resistance in U-2 OS [42,56]. To expand upon these findings, we tested whether cisplatin, another standard MAP therapy drug, shows similar stretch-modulated behavior, and whether TRPV1 mediates these effects. Pre-stretched cells were treated with cisplatin or doxorubicin in the presence or absence of AMG. As shown in Figure 1E, cyclic stretch induced resistance in U-2 OS and increased cisplatin sensitivity in SAOS-2. These effects were partially abolished in the more aggressive U-2 OS cell model by TRPV1 inhibition, specifically by 62% at 30 μM cisplatin and by 76% at 60 μM cisplatin (*p* < 0.05). In contrast, in the less aggressive SAOS-2 model, TRPV1 inhibition completely abrogated the mechanically induced changes (Figure 1E, *p* < 0.05).

A similar trend was observed for the doxorubicin-induced cell death (Figure 1F, *p* < 0.05). Importantly, TRPV1 inhibition restored drug response to baseline levels across all tested concentrations in both cell lines. (i) In U-2 OS cells, TRPV1 blockade completely reversed stretch-induced chemoresistance (Figure 1F, *p* < 0.05). (ii) In SAOS-2 cells, TRPV1 inhibition abrogated the stretch-induced increase in drug sensitivity, reducing the effect by 94% at 5 μM and by 80% at 20 μM doxorubicin (*p* < 0.05), suggesting that TRPV1 mediates this sensitization.

### 2.2. Uniaxial Stretch-Induced U-2 OS Cell Reorientation: A TRPV1-Dependent Process

In this study, we investigated the nuclear orientation of three different cell types under static conditions and after mechanical stimulation. Since the nuclear orientation closely reflects overall cell alignment, we quantified it using an ellipse-fitting method applied to fluorescently stained nuclei (see Materials and Methods and Appendix A).

Figure 2A shows a representative image of a U-2 OS cell, in which the cytoskeleton was stained fluorescent green with β-tubulin and the nucleus was counterstained blue with Hoechst dye.

To assess orientation, the angle between each nucleus’s major axis and the x-axis (corresponding to the direction of uniaxial cyclic stretch) was measured, as illustrated in Figure 2B. These angles (φ) were derived from the fitted ellipse’s major axis and subsequently transformed into the orientation parameter cos(2φ). This transformation enabled classification of nuclear alignment: values in the range −1 ≤ cos(2φ) ≤ −0.5 indicated perpendicular alignment, while values in the range 0.5 ≤ cos(2φ) ≤ 1 indicated parallel alignment (see Appendix A and Figure 2B).

The violin and bar plots in Figure 2C illustrate the distribution of cos(2φ) values across experimental conditions for all three cell lines. For both static and stretched SAOS-2 cells, a bimodal distribution emerged, with two peaks corresponding to perpendicular and parallel orientations. This observation supports previous findings [32], indicating that SAOS-2 cells lack a clear directional preference in response to cyclic strain. Similarly, hFOB cells exhibited random nuclear orientation under both control and stretched conditions, suggesting that 24 h cyclic stretch at 1 Hz did not influence their alignment (Figure 2C).

In contrast, U-2 OS cells displayed a distinct response. Even moderate mechanical stimulation (0.5% cyclic elongation) was sufficient to induce reorientation perpendicular to the stretch axis. As shown in Figure 2C, U-2 OS nuclei repositioned away from the direction of mechanical strain, a behavior that differed markedly from the random orientation seen under static conditions. This active reorientation may represent a protective mechanism against sustained mechanical loading.

Given that mechanosensitive ion channels play a key role in cellular mechanotransduction, we hypothesized that the TRPV1 ion channel might contribute to this orientation response. To test this, U-2 OS cells were co-treated with AMG during cyclic stretch. As shown in Figure 2D, blocking TRPV1 activity significantly impaired mechanically induced reorientation, indicating that TRPV1 activation is required for this response.

To further explore the underlying mechanisms, we assessed the role of cytoskeletal integrity and calcium signaling by co-treating cells with two bioactive compounds: 1 µM cytochalasin D, which disrupts actin polymerization, and 1 µM indomethacin, which affects calcium mobilization and focal adhesion dynamics. As shown in Figure 2D, both treatments inhibited U-2 OS reorientation in response to stretch. The combined results from AMG, cytochalasin D, and indomethacin treatments suggest that proper TRPV1 activity, intact actin networks, and calcium signaling are essential for the nuclear reorientation observed in U-2 OS cells under mechanical stimulation.

### 2.3. Nanomolar Capsaicin Activates TRPV1 to Reproduce Mechanical Phenotypes and Modulate Chemoresponse in OS Cells

To further support the evidence that TRPV1, rather than TRPA1, is primarily responsible for mediating OS cell mechanotransduction (as shown in Figure 1 and Figure 2), OS cells were treated for 24 h with capsaicin (a TRPV1 agonist) or ASP7663 (a TRPA1 agonist). Changes in nuclear morphology and the nuclear-to-cytoplasmic (N/C) ratio—assessed via Hoechst staining and confocal microscopy—were used as functional readouts to determine whether chemically induced activation could mimic the mechanically induced cellular responses (Figure 3A,B).

Given capsaicin’s well-documented, concentration-dependent effects—often described as a “double-edged sword”—we aimed to identify a safe concentration range that would not be toxic to healthy osteoblasts (hFOBs). Consistent with previous reports in the literature, micromolar concentrations of capsaicin—previously shown to exert anti-cancer effects in breast, prostate, and osteosarcoma cells [53,54]—also induced cytotoxicity in hFOB cells (Appendix A), highlighting the risk of off-target effects at higher doses and underscoring the need for careful dose optimization in therapeutic contexts [57]. In contrast, nanomolar concentrations of capsaicin (15–150 nM) did not induce cytotoxicity in hFOB cells, supporting its potential utility as a mechanomimetic tool Appendix A. This concentration range thus represents a safe window for targeted TRPV1 activation while minimizing the risk of off-target effects [58,59]. Moreover, as shown in Appendix A, capsaicin at 50 nM exhibited the greatest cytoprotective effect among all tested nanomolar concentrations when combined with cisplatin. For this reason, the intermediate concentration of 50 nM was selected for use in the present study.

As shown in Figure 3A, only capsaicin, but not ASP7663 treatment, successfully mimicked the mechanically induced phenotypeon OS cell models. In U-2 OS cells, capsaicin significantly increased nuclear area to 749.1 µm^2^ compared to 553.7 µm^2^ in untreated cells (*p* < 0.0001; mean difference ± SEM: 195.4 ± 27.7), closely resembling the effect of mechanical stimulation. A similar response was observed in SAOS-2 cells, where nuclear area increased from 612.0 µm^2^ in controls to 827.0 µm^2^ following capsaicin treatment (*p* < 0.0001; mean difference ± SEM: 215.0 ± 25.0). In contrast, ASP7663 did not produce any comparable response. Notably, in healthy osteoblasts, neither 50 nM capsaicin (*p* > 0.05; mean difference ± SEM −398.5 ± 478.3) or 50 nM ASP7663 (*p* > 0.05; mean difference ± SEM: −601.2 ± 418.7) induced detectable changes in nuclear morphology compared to controls, confirming the selective mechanosensitivity of OS cells (Appendix A).

Confocal microscopy analysis (Figure 3B) further confirmed that TRPV1 activation by capsaicin reproduced key aspects of the mechanical response in both U-2 OS and SAOS-2 cells. In U-2 OS cells, capsaicin treatment increased the nuclear-to-cytoplasmic (N/C) ratio from 0.26611 to 0.4376 (*p* < 0.0001; difference between means ± SEM: 0.1716 ± 0.01245). In contrast, SAOS-2 cells exhibited a modest but significant decrease in the N/C ratio following capsaicin treatment, from 0.23349 to 0.19188 (*p* < 0.001; difference between means ± SEM: −0.04160 ± 0.01126).

Capsaicin also promoted cell migration, as demonstrated by wound healing assays (Figure 3C), with a 1.3-fold increase in U-2 OS and a 1.4-fold increase in SAOS-2 cells (*p* < 0.01), further mirroring the mechanically induced phenotype. Adhesion assays revealed that TRPV1 activation by capsaicin similarly reproduced stretch-induced changes, leading to a 1.4-fold increase in adhesion in U-2 OS cells (*p* < 0.0001) and a 1.5-fold decrease in SAOS-2 cells (*p* < 0.01), indicating a cell line-specific regulatory effect (Figure 3D).

To further confirm that these effects were specifically mediated by TRPV1, all experiments were repeated with co-treatment using AMG. Strikingly, AMG abolished all capsaicin-induced changes (Figure 3A–D, *p* < 0.05), providing strong evidence that TRPV1 activation is a key transducer of both mechanical and chemical stimuli in OS cell biology.

To assess the translational relevance of capsaicin as a chemotherapeutic adjuvant at a 50 nM concentration—and given its lack of cytotoxicity and observed cytoprotective effect in hFOB cells—we next investigated whether capsaicin could modulate the efficacy of two standard MAP chemotherapy agents, cisplatin and doxorubicin, when used in combination across the three cell lines (Figure 3E–G).

As shown in Figure 3E–G, capsaicin exhibited no cytotoxicity at nanomolar concentrations (15–150 nM) in any of the tested cell lines, supporting its utility as a mechanomimetic tool. This concentration window thus represents a safe range for targeted TRPV1 activation, confirming 50 nM as a safe and effective selected dose.

Notably, a low dose of capsaicin mirrored the effects of mechanical preconditioning, altering chemotherapeutic sensitivity in both U-2 OS and SAOS-2 cell lines (Figure 3F,G). As shown in Figure 3F, chemical activation of TRPV1 with 50 nM capsaicin significantly decreased U-2 OS cell sensitivity to cisplatin, resulting in a 10% increase in cell viability compared to TRPV1-inactive controls at both 30 µM (*p* < 0.001) and 60 µM (*p* < 0.0001). A similar pattern was observed for doxorubicin treatment, where TRPV1 activation led to an approximately 15% reduction in drug-induced cytotoxicity (Figure 3F; *p* < 0.01 at 5 µM and *p* < 0.001 at 20 µM). In contrast, SAOS-2 cells exhibited increased chemosensitivity following TRPV1 activation. Capsaicin treatment enhanced cisplatin efficacy, resulting in at least a 10% decrease in cell viability at both 30 µM (*p* < 0.01) and 60 µM (*p* < 0.05), and improved doxorubicin sensitivity by approximately 5% at 5 µM and 20 µM (both *p* < 0.01; Figure 3G).

To further dissect the role of TRPV1 in this context, we also evaluated the effects of its pharmacological inhibition using AMG9810. Importantly, co-treatment with AMG abolished the capsaicin-induced changes in cisplatin and doxorubicin sensitivity in both OS cells (Figure 3F,G), confirming that these effects are specifically mediated through TRPV1 activation.

Given that TRPV1 inhibition improved chemotherapy efficacy in U-2 OS-like cells, we next explored whether AMG alone, or in combination with cisplatin, could be cytotoxic to hFOBs. As shown in Appendix A, AMG treatment had no significant impact hFOB viability, indicating that TRPV1 blockade does not introduce additional toxicity in non-malignant cells.

Taken together, these results support the selective use of TRPV1 modulators as adjuvant agents. Capsaicin may offer a dual benefit in SAOS-2-like osteosarcoma by enhancing chemotherapeutic efficacy while protecting healthy bone cells. In contrast, TRPV1 inhibition with AMG could improve treatment outcomes in U-2 OS-like tumors without compromising the viability of healthy osteoblasts.

### 2.4. Chemical Activation of TRPV1 Replicate Mechanical Signals Governing Osteosarcoma Cell Adhesiveness and Edge Complexity

As described above, TRPV1 activation—whether mechanical or chemical (Figure 1B and Figure 3D)—acts as a central regulator of adhesion dynamics and underlies the divergent adhesion responses observed between the two OS subtypes. Based on this, we hypothesized that TRPV1 signaling may differentially influence cell edge architecture, a key feature involved in substrate engagement. Previously, we reported that mechanical stretch does not change overall cell height or surface roughness but selectively increases peripheral roughness in SAOS-2 cells [32]; Appendix A. In the present study, we extended this analysis to investigate whether TRPV1 activation by capsaicin produces similar morphological effects. To this end, we employed contact-mode atomic force microscopy (AFM) combined with box-counting fractal dimension (FD) analysis—a method well-suited for quantifying the complexity of irregular, self-similar structures such as cell boundaries [60,61,62,63]; see Appendix A, Appendix A. Figure 4A shows the AFM-analyzed cell regions, which were deliberately restricted to areas exhibiting peripheral roughness and protrusion complexity, highlighted in purple and green, respectively.

Consistent with our previous findings, capsaicin treatment did not significantly affect overall cell height or global surface roughness in either U-2 OS or SAOS-2 cells (Appendix A). However, analysis of peripheral roughness revealed opposite trends: a mild, non-significant increase in U-2 OS cells and a decrease in SAOS-2 cells following TRPV1 activation (Figure 4B). While these changes did not reach statistical significance, they suggest a potential cell type-specific response in edge architecture linked to TRPV1-mediated signaling.

To more precisely quantify edge complexity, we performed FD analysis on high-resolution AFM images of OS cells cultured on rigid glass substrates (Figure 4C). This approach enabled nanoscale assessment of boundary irregularities—such as lamellipodia-like protrusions—that cannot be accurately visualized on flexible silicone due to background deformation and the absence of a stable imaging plane. FD analysis revealed that capsaicin had no significant effect on SAOS-2 cells (control: 1.051 ± 0.052; capsaicin: 1.029 ± 0.028; *p* > 0.05.). In contrast, U-2 OS cells displayed a marked increase in FD following TRPV1 activation (control: 1.103 ± 0.06; capsaicin: 1.272 ± 0.09; *p* < 10^−6^), which was fully reversed by co-treatment with the TRPV1 antagonist AMG (1.045 ± 0.05; *p* < 0.0001; Figure 4C). Together, these findings demonstrate that TRPV1 activation significantly enhances protrusion complexity in U-2 OS cells—a hallmark of migratory and adhesive phenotypes—while SAOS-2 cells remain morphologically unresponsive. This highlights a cell line-specific role for TRPV1 in regulating morphological plasticity at the nanoscale level.

As previously noted, the use of silicone substrates limited direct high-resolution visualization of cellular protrusions. To address this, we employed complementary indirect approaches to assess whether mechanical activation of TRPV1 induces cytoskeletal structures similar to those observed following capsaicin stimulation. Given our earlier findings of enhanced adhesion and motility under both conditions, we hypothesized that lamellipodia-like protrusions—key drivers of these behaviors—might be involved.

To test this, we stained filamentous actin (F-actin) with phalloidin and performed immunofluorescence imaging (Figure 4D). Both mechanically stretched and capsaicin-treated U-2 OS cells exhibited accumulation of peripheral F-actin, consistent with lamellipodia formation. Although these observations are qualitative, the effect was clearly abolished by co-treatment with AMG (Figure 4D), suggesting that TRPV1 activation—whether mechanical or chemical—drives cytoskeletal remodeling at the leading edge.

To further support these findings, we conducted a functional assay to evaluate whether TRPV1-induced protrusions contribute not only to adhesion formation but also to the stability of adhesive interactions. Specifically, we performed a cell detachment assay to measure resistance to shear-induced dissociation. In U-2 OS cells, mechanical stimulation led to an approximately 65% reduction in cell detachment compared to static conditions (*p* < 0.001; Figure 4E). A similar effect was observed following chemical activation of TRPV1 with 50 nM capsaicin, which resulted in a ~50% decrease in detachment relative to untreated controls (*p* < 0.01; Figure 4E). Notably, both effects were completely abolished by co-treatment with the antagonist AMG, further supporting the role of TRPV1 signaling in enhancing the stability of adhesive interactions (Figure 4E).

### 2.5. TRPV1 Activation Mediates Nuclear Localization of Src and Triggers Histone 3 Acetylation

Given our previous findings that the Src—H3 acetylation axis plays a critical role in tumor cell mechanosignaling—serving as a key transduction pathway in response to 1 Hz mechanical stimulation of OS cells [42]—we hypothesized that TRPV1 activation might be a component of this signaling cascade. To investigate this, we used Western blot analysis to assess whether the AMG inhibitor could abrogate the mechanically induced changes in SRC and acetylated histone H3 (acH3) protein levels.

We compared protein expression in stretched OS cells treated with or without AMG to further dissect the role of TRPV1 in mechanotransduction. Remarkably, inhibition of TRPV1 alone was sufficient to replicate the broad inhibitory effects observed with GsMTx4, a general MS channel blocker [42]. Consistent with prior results, the two OS cell lines (U-2 OS and SAOS-2) displayed opposing responses to mechanical stimulation while Src_H3 acetylation axis. Western blot analysis revealed that AMG treatment abolished the mechanically induced changes in Src expression in both models (Figure 5A). Specifically, AMG reversed the 2-fold upregulation of Src in U-2 OS cells (*p* < 0.001) and neutralized the 0.5-fold increase in SAOS-2 cells (*p* < 0.01), confirming a cell line-specific role for TRPV1 in Src modulation.

To further validate TRPV1′s role in regulating downstream epigenetic modifications, we next assessed levels of acetylated histone 3 levels. As shown in Figure 5B, AMG treatment fully reversed the stretch-induced elevation in acH3 levels, restoring them to baseline in both cell lines (*p* < 0.0001 for U-2 OS; *p* < 0.01 for SAOS-2). Together, these findings confirm that TRPV1 activity is essential for the mechanical regulation of the Src–acH3 signaling axis in OS cells, exerting opposing regulatory effects in the two OS cell lines—upregulating the Src–acH3 axis in U-2 OS cells while downregulating it in SAOS-2 cells.

Next, to determine whether pharmacological modulation of TRPV1 could directly regulate the mechanosensitive Src–acH3 axis independently of mechanical stimulation, we examined the effects of TRPV1 activation and inhibition in OS cells under static conditions (i.e., without 1 Hz stimulation). Cells were treated with capsaicin alone or in combination with AMG, and protein levels were analyzed by Western blot.

Quantitative analysis revealed that pharmacological activation or inhibition of TRPV1 was sufficient to modulate the Src–acH3 axis in both OS cell lines, even in the absence of mechanical input. Specifically, TRPV1 activation by capsaicin resulted in a significant ~30% increase in Src protein levels in U-2 OS cells (Figure 5C, *p* < 0.001), following a trend similar to that observed with mechanical stimulation (Figure 5A). Consistent with these findings, densitometric analysis of chemically induced acH3 levels (Figure 5D) revealed patterns that mirrored those observed under mechanical stimulation (Figure 5B) in both OS cell lines. In U-2 OS cells, capsaicin treatment alone led to a 2.5-fold increase in acetylated histone H3 levels (Figure 5D, *p* < 0.01), which was notably greater than the increase observed under mechanical stimulation (Figure 5B). In contrast, in SAOS-2 cells, TRPV1 activation—whether mechanical or chemical—resulted in a comparable 1.5-fold decrease in acH3 levels (Figure 5B,D).

Importantly, these divergent effects were significantly attenuated by co-treatment with the TRPV1 antagonist AMG, supporting the specificity of TRPV1 involvement. In U-2 OS cells, AMG co-treatment significantly reduced both Src and acH3 responses (*p* < 0.01), while in SAOS-2 cells, the capsaicin-induced changes were significant only for in acH3 (*p* < 0.05) (Figure 5A–C).

Although the overall patterns closely paralleled those observed with mechanical modulation of the Src–acH3 axis, the magnitude of Src changes induced by pharmacological activation was generally less pronounced. Given that Src nuclear localization—dynamically regulated by mechanical cues—has been proposed as prognostic marker of OS malignancy [35,36,45], we hypothesized that Src kinase may exert distinct regulatory effects on this shared signaling axis, depending on its subcellular localization and the specific cellular context.

To test whether TRPV1 activity influences the subcellular localization of Src, we performed immunofluorescence staining in both U-2 OS and SAOS-2 cells under conditions of TRPV1 activation—either mechanical or chemical—as well as pharmacological inhibition (Appendix A). Nuclear Src levels were quantified using the Cell Profiler pipeline, as outlined in Appendix A. Comparative analysis of corrected total cell fluorescence (CTCF) revealed that mechanical activation of TRPV1 significantly increased nuclear localization of Src in U-2 OS cells by nearly twofold (*p* < 0.0001), whereas mechanically stretched SAOS-2 cells showed a significant 1,4-fold reduction in nuclear Src levels (*p* < 0.01; Figure 5E). Importantly, these changes were reversed by AMG antagonist, with nuclear Src localization returning to baseline in both U-2 OS (*p* > 0.0001) and SAOS-2 cells (*p* < 0.05) (Figure 5E).

Figure 5F demonstrates that chemical activation of TRPV1 reproduces the same Src nuclear relocalization trend observed with mechanical regulation (Figure 5E), even in the absence of any mechanical stretch. The induced changes were modest in the U-2 OS model, but more pronounced in SAOS-2 cells. In capsaicin-treated U-2 OS cells, Src nuclear localization increased by approximately 1.3-fold (*p* < 0.001), while in SAOS-2 cells, capsaicin induced a clear redistribution of Src away from the nucleus, decreasing nuclear localization by approximately 2.3-fold (*p* < 0.0001). Again, co-treatment with AMG reversed these effects, restoring Src localization in both U-2 OS (*p* < 0.05) and SAOS-2 cells (*p* < 0.0001) (Figure 5F).

Under basal physiological conditions, inactive Src typically resides in the cytoplasm or perinuclear region, as reported in both cancerous and non-cancerous cells [36,47]. To determine whether TRPV1-mediated Src nuclear localization is specific to malignant cells, we performed the same CTCF analysis in non-malignant hFOB osteoblasts treated with either 1 Hz cyclic stretch or 50 nM capsaicin (Appendix A).

In hFOB non-malignant (healthy) model, neither mechanical nor chemical activation of TRPV1 induced detectable nuclear translocation of Src compared to untreated controls (Figure 5G,H). Western blot analysis further confirmed that total Src protein levels remained unchanged following either form of stimulation (Figure 5G,H), reinforcing the notion that TRPV1-mediated modulation of Src is specific to the osteosarcoma context. Notably, in hFOB cells, Src remained excluded from the nucleus and was not influenced by either mechanical or chemical TRPV1 activation (Figure 5G,H). Notably, and in line with previous findings [43,45], Src was consistently excluded from the nucleus in hFOB cells and remained unaffected by TRPV1 activation, regardless of the applied stimulus (Figure 5E,F). Given Src integration within the cell’s tensegrity-based mechanical architecture [5], it is plausible that mechanical stretch could trigger Src’s subcellular relocalization via cytoskeletal force transmission.

Collectively, these findings underscore TRPV1′s role as a multimodal sensor that, even in the absence of mechanical input, acts as a molecular switch to engage the Src–acH3 signaling axis. TRPV1 activation appears central to fine-tuning epigenetic regulation by modulating Src subcellular localization, thereby contributing to the phenotypic divergence observed between osteosarcoma subtypes. This cell type-specific mechanism may reflect a broader paradigm in which mechanosensitive channels integrate both mechanical and chemical cues to regulate gene expression in cancer cells. A schematic of the proposed molecular mechanism is illustrated in Figure 6.

## 3. Discussion

Osteosarcoma, the most common primary malignant bone tumor, posed a formidable clinical challenge due to its profound heterogeneity, complex microenvironment, and resistance to conventional therapies [64,65]. Despite decades of combined surgical and chemotherapeutic strategies, survival rates plateaued, and the presence of metastatic disease at diagnosis continued to confer poor prognosis [66,67]. While significant progress had been made in delineating molecular subtypes and identifying therapeutic targets through genomic profiling and molecular classification [68,69], the biomechanical properties of the osteosarcoma cells remained underexplored.

Growing evidence suggested that mechanical cues—such as matrix stiffness, cellular adhesion forces, and extracellular vesicle-mediated mechanotransduction—played a critical role in tumor progression, invasion, and therapeutic resistance [64]. Understanding how osteosarcoma cells perceive and respond to mechanical stimuli was expected to uncover novel vulnerabilities, thereby complementing existing molecular and immunological treatment strategies [67,70].

Given that alterations in cellular mechanoproperties have been linked to changes in morphology, migration, adhesion dynamics, proliferation, and alignment [71,72], we employed uniaxial cyclic stretching to investigate how osteosarcoma cells reprogram normal mechanosensing pathways to support aggressive behavior. When cyclic stretch is applied uniaxially—that is, along a single axis—cells undergo strain aligned with the direction of force. Their adaptive responses are highly cell type-specific: while some cells align with the direction of stretch, others reorient perpendicularly to minimize mechanical stress. A well-documented response to cyclic stretch in many adherent cell types was the reorientation of the actin cytoskeleton and cellular long axis nearly perpendicular to the direction of stretch, a process shown to be Src-dependent [73].

In our previous study, we observed that SAOS-2 cells failed to reorient within 24 h of cyclic stretch exposure, maintaining their original alignment [32]. By contrast, others reported that U-2 OS cells reoriented perpendicularly, reorganizing their cytoskeleton to escape sustained strain [39]. In line with these findings, the present study demonstrates that U-2 OS cells exhibit a remarkable capacity for reorientation, responding even to mild cyclic stretch (0.5% elongation). Moreover, in line with date reported in the literature [74,75,76], the present study indicated that this reorientation depended on proper TRPV1 activation and was indirectly reliant on actin filament remodeling and calcium-dependent signaling, as evidenced by the effects of AMG 9810, cytochalasin D, and indomethacin treatments.

TRPV1, also known as the capsaicin receptor, thermal receptor, or pain receptor, is a non-selective cation channel that shares features with many MS channels. It has been distinguished by its role as a true multisensor, capable of responding to a broad spectrum of endogenous stimuli—including mechanical forces, heat, low pH, lipid metabolites, and inflammatory mediators [77]. Although initially characterized in the context of pain perception and inflammation [78], TRPV1 had increasingly been implicated in cancer cell adaptation to mechanical and oxidative stress [25]. Mechanical activation of TRPV1 was shown to trigger downstream signaling cascades that regulate key oncogenic processes such as proliferation, migration, invasion, and therapy resistance [18,19,20,21,22]. Understanding how OS cells respond to mechanical stimuli through TRPV1 and other channels—both in vitro and in vivo—may uncover early markers of aggressiveness, such as stretch reorientation, adhesion changes, or cytoskeletal remodeling. These traits could be used in biophysical screening tools (e.g., microfluidic devices or mechanosensitive biosensors) for early detection or tumor stratification. Although data remain limited, variations in TRPV1 expression or activity across OS models may indicate mechanotransductive heterogeneity, which could support the development of biomechanical-based diagnostic or prognostic tools.

In this study, we demonstrated that TRPV1—reported to be comparably expressed across the two OS cell lines in the Human Protein Atlas database of cancer cells (https://www.proteinatlas.org (accessed on 6 April 2025))—functions as a central node in the mechanotransduction machinery of OS cells, while showing no apparent role in mechanoregulation in healthy hFOB osteoblasts, for which no comparable expression data are available. Specifically, TRPV1 activity played a prominent role in regulating both morphological adaptation and drug responsiveness under mechanical stimulation in OS cells. In the present study, we found that our experimental conditions successfully chemically mimicked the TRPV1-mediated responses typically induced by mechanical stretch. Capsaicin, a well-studied nutraceutical compound [49,50,51,52], and its selective TRPV1 antagonist AMG9810, extensively used in preclinical models of pain, inflammation, and cancer [25,79], were both effective in modulating TRPV1 activity in OS cells while sparing healthy hFOB osteoblasts.

By treating cells with nanomolar concentrations of capsaicin (50 nM), we chemically activated TRPV1 while avoiding the off-target effects and cytotoxicity often observed at micromolar levels [53,54,55]. In parallel, AMG9810 at 560 nM, preincubated to outcompete the agonist, effectively antagonized TRPV1 under the same conditions. This study demonstrated that chemical and mechanical activation of TRPV1 elicited similar phenotypic responses in OS cell models. Interestingly, both compounds effectively modulated TRPV1-dependent pathways in U-2 OS and SAOS-2 cells, while sparing hFOB cells, underscoring their selectivity and highlighting the therapeutic potential of targeting TRPV1 in OS without compromising normal bone cell integrity.

This study demonstrated that both chemical and mechanical activation of TRPV1 induced comparable responses in the OS models tested, notably increasing nuclear dimensions and migration rate. Interestingly, our results showed that TRPV1 activation elicited divergent effects on the nuclear-to-cytoplasmic (N/C) ratio and adhesion dynamics in SAOS-2 and U-2 OS cells—two features previously associated with metastatic potential [80,81].

Ultrastructural analysis using AFM, and complemented by fractal dimension analysis—a powerful quantitative method previously employed to distinguish between healthy and cancerous cervical epithelial cells [60], and to evaluate the effects of actin polymerization inhibitors on cell boundary roughness in human epithelial cells [61]—revealed distinct differences in surface complexity and peripheral architecture between the two OS subtypes. Supporting these findings, immunofluorescence microscopy and detachment assays demonstrated that TRPV1 activation in U-2 OS cells promoted the formation of lamellipodia-like, actin-rich protrusions and enhanced initial adhesion, which are features associated with increased adhesion plasticity. In contrast, SAOS-2 cells exhibited minimal morphological remodeling, highlighting subtype-specific differences in mechanosensitivity, likely reflecting underlying intrinsic molecular variations [82]. Although further investigation is needed, the differing mechanosensitive responses between U-2 OS and SAOS-2 cells might be linked to intrinsic differences in cytoskeletal regulation. Specifically, changes in Rho GTPases and actin-binding protein expression or activity have been suggested to affect actin remodeling and mechanotransduction [83].

At the molecular level, our data demonstrated that the Src–histone H3 acetylation (acH3) mechanosignaling axis, previously identified by our group and others as a key mediator of cellular mechanosensation [44], was tightly regulated by TRPV1 activation—regardless of the activation method (mechanical or chemical).

Histone H3 acetylation and Src expression are known to regulate gene expression, cell signaling, and cancer progression [84], while aberrant acetylation and elevated Src levels are associated with enhanced tumorigenicity [85,86]. Notably, we found that TRPV1 functions as a molecular switch, exhibiting subtype-specific modulation: it upregulates the Src–acH3 axis in U-2 OS cells, whereas it downregulates the same pathway in SAOS-2 cells.

Notably, we found that TRPV1 acted as a molecular switch, upregulating the Src–acH3 axis in U-2 OS cells, while downregulating it in SAOS-2 cells. Given that Src interacts with lamin A/C [45], a key structural protein of the nuclear envelope [35,87], and has been previously detected in the nucleus of OS cells [35,36], our findings suggest that TRPV1 modulated Src subcellular localization in a subtype-dependent manner. Specifically, TRPV1 activation either promoted nuclear translocation or induced cytoplasmic extrusion of Src, with minimal impact on hFOB cells. This subtype-specific spatial regulation of Src supports the hypothesis that TRPV1 fine-tunes epigenetic and transcriptional responses through localization-dependent control of Src, thereby contributing to the mechanobiological heterogeneity across OS subtypes.

The differential TRPV1 responsiveness observed between U-2 OS and SAOS-2 cells may be related to their distinct differentiation states, which could, in turn, influence cytoskeletal organization and modulate TRPV1-mediated mechanotransduction, as well as downstream pathways such as the Src–H3 acetylation axis. However, these potential links remain to be clarified through targeted experimental studies.

In addition to their role in mechanotransduction, phenotypic differences and subtype-specific molecular features might hold promise as early indicators of osteosarcoma aggressiveness, potentially offering avenues for diagnostic stratification and therapeutic targeting. Nevertheless, further investigation is needed to assess their clinical relevance and applicability.

## 4. Materials and Methods

### 4.1. Cell Culture and Chemical Materials

Osteosarcoma cell lines Human SAOS-2 (HTB-85) and U-2 OS (HTB-96) and human fetal osteoblast cell line hFOB (CRL-3602) were purchased from the American Type Culture Collection (Rockville, MD, USA). The osteosarcoma cells were cultured in Dulbecco’s Modified Eagle’s Medium (4.5 g/L glucose) (DMEM), while 1.19 hFOB were cultured in Ham’s F-12 Nutrient Mixture (F-12) medium (1:1) (Gibco, Life Technologies, Carlsbad, CA, USA). They were both supplemented with 10% fetal bovine serum (FBS) (Euroclone s.p.a., Milano, Italy), Penicillin–Streptomycin Solution 100X (Gibco, Life Technologies, Carlsbad, CA, USA), and Amphotericin B 100X (Biowest, Riverside, MO, USA) within cell culture flasks at 37 °C in an atmosphere of 5% CO_2_. Rat type I collagen (Enzo Life Sciences, Farmingdale, NY, USA) was used for coating cell plates. The culture medium was changed twice a week, during which non-adherent cells were discarded.

### 4.2. Mechanical Stretch Application

Cells were cultured on deformable silicone plates (CellScale Biomaterials Testing) and subjected to cyclic uniaxial stretch (0.5% elongation, 4830 µε) along the x-axis using the MechanoCulture FX device at 1 Hz for 24 h, following a 1 h stretch/3 h rest cycle as previously described [32,42]. Two plates were prepared per experiment: one stretched and one static control. Stretching was also performed in the presence of specific ion channel blockers. Analyses were conducted immediately post-stimulation either on-plate or on harvested cells; lasting effects were assessed on reseeded cells on glass (Appendix A). Fluorescence and spectrometry measurements on plates were performed using a custom tray for optimal positioning on a TECAN Spark microplate reader [32].

### 4.3. Chemical Activation and Inhibition of MS Channels Through Soluble Compounds

To assess whether chemical activation could replicate the effects of the stretch stimulation of the MS ion channel, each osteosarcoma cell line was seeded on collagen precoated conventional tissue culture supports and then incubated under conventional culturing conditions with either 50 nM Capsaicin, a selective agonist of TRPV1 channel (Y0000671, Sigma-Aldrich Chemical Co., St. Louis, MO, USA) or 50 nM ASP7663, selective activator of the TRPA1 channel (SML1467, Sigma-Aldrich Chemical Co., St. Louis, MO, USA) for 24 h. Cells were seeded at different densities according to the assay: density of 1.5 × 10^4^ cells per well in 24-well plates for confocal and fluorescence microscopy, 400 cells per well in 96-well plates for Adhesion Assay, 1.5 × 10^4^ cells per well in 96-well plates for Wound Healing Assay, and 220 cells/mm^2^ cells per well in 96-well plates for Cytotoxicity Assay. Cotreatment experiments were conducted by repeating all procedures following a 30 min preincubation with 560 nM AMG9810 (AMG), a selective and competitive TRPV1 antagonist (S6934, Selleck Chemicals, Houston, TX, USA), in both U-2 OS and SAOS-2 cell lines. AMG9810, a potent aryl cinnamide-class inhibitor (IC_50_ in the low nanomolar range; [79]), was used at 560 nM—approximately 10 times higher than the capsaicin dose—to ensure sustained and effective blockade of both chemical and mechanical TRPV1 activation.

### 4.4. Cell Count, and Protein Extraction and Quantification

Immediately after treatment, both TRPV1-activated and control cells were trypsinized and counted using two automated, chip-based systems: the TECAN Spark multimode reader (Tecan Group, Männedorf, Switzerland) and the NucleoCounter^®^ NC-200^®^ automated cell counter, employing the Via1-Cassette™ cell sampling and staining cartridge (ChemoMetec A/S, Allerød, Denmark). Following counting, cells were centrifuged and washed with PBS. Histones were isolated from the pellet via acid extraction as described by us [42]: pellets were incubated overnight in 0.2 N HCl at 4 °C, centrifuged, and the supernatant neutralized with 2 M NaOH. Protein concentrations were measured by Bradford assay and DC Protein Assay (BioRad) using BSA standards, with absorbance read at 595 nm on the TECAN Spark.

### 4.5. Cell Viability Assay and Cytotoxicity Assay

To determine drug efficacy, the effect of mechanical or chemical pre-treatment on doxorubicin- or cisplatin-induced cytotoxicity was assessed using an MTT assay (Merk Life Science, Milano, Italy) as described previously [42,56] (Appendix A). Cells were seeded on silicone plates at a density of 220 cells/mm^2^, allowed to adhere, and then stretched cyclically for 24 h in the presence or absence of 560 nM AMG9810. For the assays conducted on conventional supports, cells were seeded at the same density on 96-well plates, allowed to adhere, and then pretreated with 50 nM Capsaicin for 24 h. They were then treated for 24 h with serum-free medium containing doxorubicin (0–20 μM) or cisplatin (0–60 μM). After treatment, 20 μL of MTT solution (5 mg/mL in PBS with Ca^2+^ and Mg^2+^) was added to each well and incubated at 37 °C with 5% CO_2_ for 2 h. Formazan crystals were dissolved by adding 100 μL of extraction buffer (5% SDS in N,N-dimethylformamide), followed by another 2 h incubation under the same conditions. Absorbance was measured at 570 nm using a Tecan Infinite^®^200 PRO reader. Cell viability was calculated as the ratio of absorbance in treated wells to control (untreated) wells. Experiments included three biological replicates with four technical replicates per condition.

### 4.6. Confocal Microscopy

Confocal images were acquired from cells seeded at a density of 110 cells/mm^2^ on rat type I collagen-coated (50 µg/mL) silicone plates or conventional glass coverslips, as previously described [32]. Following mechanical or chemical stimulation, cells were fixed in 4% paraformaldehyde for 20 min at room temperature, then rinsed with ultrapure water.

Imaging was performed using an Olympus LEXT OLS 4000 confocal microscope equipped with a 405 nm laser and ×20 (NA 0.60) or ×50 (NA 0.95) objectives. Images were captured over areas of 648 µm × 648 µm or 258 µm × 258 µm, respectively, at a resolution of 4096 × 4096 pixels (~0.025 µm/pixel). All images were exported as TIFF files for subsequent analysis.

### 4.7. Immunofluorescence Microscopy

After mechanical or biochemical activation of the TRPV1 channel—with or without the antagonist AMG9810—cells were washed with PBS and fixed with 4% paraformaldehyde for 20 min at room temperature. Following an additional PBS wash, cells were permeabilized with 0.1% Triton X-100 in PBS for 15 min, then blocked with 10% donkey serum in PBS for 1 h at room temperature. Samples were subsequently incubated overnight at 4 °C with the following primary antibodies, diluted in 0.1% BSA in PBS:SRC: (1:100, #sc-32789, Santa Cruz Biotechnology, TX, USA).β-Tubulin: (1:100, #GTX101279, GeneText Irvine, CA, USA).

The next day, glass coverslips and silicone wells were washed with PBS and incubated for 1 h at room temperature with Hoechst 33342 (1:600, #33342, Sigma-Aldrich, St. Louis, MO, USA), along with the appropriate secondary antibodies diluted in 0.1% BSA in PBS:Alexa Fluor 488–conjugated anti-rabbit IgG (1:400, #A21206, Thermo Fisher Scientific, MA, USA).Alexa Fluor 568–conjugated anti-mouse IgG (1:400, #A10037, Thermo Fisher Scientific, MA, USA).

To visualize peripheral actin structures in U-2 OS cells, samples were also stained with Phalloidin–iFluor 488 (1:1000, #ab176753, Abcam, MA, USA).

After final washes in PBS, samples were mounted using ProLong™ Gold Antifade Mountant (Thermo Fisher Scientific, MA, USA). Fluorescence imaging was performed using a Zeiss Axioplan 2 microscope at 20×, 40×, and 100× magnifications.

### 4.8. Microscopy Image Analyses

Confocal and immunofluorescence images were analyzed using ImageJ software (NIH, Bethesda, MD, USA) to quantitatively compare morphological differences between control and treated samples (Appendix A). Nuclear size was measured from Hoechst-stained immunofluorescence images, following the protocol described by us [42]. Images were binarized, and the Fit Ellipse function in ImageJ was used to determine the minimum enclosing ellipse for each nucleus. At least 30 cells per condition were analyzed across three biological replicates.

For nuclear-to-cytoplasmic (N/C) ratio analysis, confocal images were used. The total cell area was manually outlined using the freehand selection tool, and the nuclear area was determined via elliptical selection. Measurements were conducted on at least 27 cells per condition across 3 biological replicates.

### 4.9. Cell Re-Orienteering Along X Stretching Axis

To investigate the effect of uniaxial cyclic stretch on cell reorientation across different cell types, we subjected cells to mechanical strain at 1 Hz for 24 h. Three cell types were analyzed: SAOS-2, U-2 OS, and hFOB. Cell nuclei were fluorescently stained, and their orientation was quantified using an ellipse-fitting method (Appendix A).

The orientation angle (φ) was defined as the angle between the major axis of each fitted ellipse and the direction of uniaxial stretch (i.e., the x-axis). These angles were then transformed into the orientation parameter cos(2φ), which enabled classification of nuclear alignment. Perpendicular orientation was defined within the range −1 ≤ cos(2φ) ≤ −0.5, and parallel orientation within 0.5 ≤ cos(2φ) ≤ 1 Appendix A.

To evaluate the involvement of TRPV1 mechanosensitive ion channels, U-2 OS cells were co-treated with the specific TRPV1 antagonist AMG9810 during stretch stimulation. Additionally, to investigate the cytoskeletal structure involved in nuclear reorientation process, U-2 OS cells were treated with either 1 µM cytochalasin D (to disrupt actin polymerization) or 1 µM indomethacin (to interfere with calcium mobilization and focal adhesion formation).

### 4.10. Adhesion Assay

Cell adhesion following mechanical or biochemical activation of the TRPV1 channel was assessed as previously described ([32]; Appendix A). Pre-treated cells, with or without co-treatment with 560 nM AMG9810, were trypsinized, counted, and seeded into 96-well plates pre-coated with rat type I collagen at a density of 400 cells/well in DMEM high glucose/Ham’s F-12 (1:1) medium.

After 24 h of incubation at 37 °C under standard culture conditions, wells were gently washed and stained with 0.5% crystal violet in 20% methanol (Sigma-Aldrich, St. Louis, MO, USA) for 10 min. Excess dye was removed through thorough washing with water.

Adherent cells were imaged using an Olympus CKX53 inverted microscope equipped with an EP50 digital camera. Adhesion was quantified by counting stained cells in 10 randomly selected fields per well at 4× magnification using ImageJ software (NIH, Bethesda, MD, USA). Data represent the mean of three independent experiments, each performed with six technical replicates per condition.

### 4.11. Detachment Assay

The detachment assay was performed as described in [88]. Briefly, SAOS-2 or U-2 OS cells were seeded and treated as outlined in the adhesion assay. After 24 h of incubation, initial images were captured. To induce detachment, the plates were incubated for 1 h at 37 °C with vigorous shaking (240 rpm). The remaining attached cells were then stained with crystal violet as described above, and final images were taken. Detachment was quantified as the percentage of remaining attached cells relative to the initial cell number.

### 4.12. Western Blotting Analysis

Following mechanical stimulation, treated cells and their counterparts were trypsinized, pelleted, and their protein content was quantified as described above (Appendix A). Total protein extracts (15–30 µg/lane), or histone acid extracts (1–3 µg/lane), were resolved on 4–20% Mini-PROTEAN^®^ TGX™ precast gels (BioRad, Hercules, CA, USA) as described in [42]. Proteins were transferred onto PVDF membranes (Amersham, UK), blocked at room temperature with either 5% non-fat milk for 2 h or EveryBlot Blocking Buffer (BioRad) for 5 min, washed, and then incubated with primary and HRP-conjugated secondary antibodies. Blots were developed using an Enhanced Chemiluminescence (ECL) system (Amersham, UK). The following primary antibodies were used:GAPDH (1:10,000, GTX100118, GeneTex, Irvine, CA, USA).Src (1:500, #2108, Cell Signaling Technology, Danvers, MA, USA).Histone H3 (1:2000, ab1791, Abcam Cambridge, UK).Acetyl-Histone H3 (Lys9/Lys14) (1:500, #9677, Cell Signaling Technology, Danvers, MA, USA).

Band intensities were quantified in ImageJ (NIH, Bethesda, MD, USA) and expressed in arbitrary units (AU). Data were analyzed using GraphPad Prism 9 (version 9.0.0), based on three independent experiments.

### 4.13. Wound Healing Assay (Scratch Assay)

Chemically or mechanically activated cells (with or without mechanosensitive channel inhibitor) were compared for their migrative capacity as previously described [42]. Pretreated cells and their counterparts were seeded into collagen-pre-coated 96-well plates at 1 × 10^4^ cells/well (Appendix A). After 24 h of incubation at 37 °C with 5% CO_2_, confluent monolayers were scratched using a 10 µL pipette tip, and wells were rinsed with PBS to remove detached cells, as previously described. Cells were then cultured in serum-free DMEM, with or without 560 nM AMG9810, and incubated for another 24 h.

Scratch closure was imaged at 0 h and 24 h using an Olympus CKX53 microscope with an EP50 digital camera. Wound closure percentage (WC%) and relative wound density (RWD) were quantified using ImageJ (NIH, USA) and a Spark multi-mode plate reader (Tecan Group Ltd., Männedorf, Switzerland), respectively. Statistical analysis was performed on three independent experiments with at least four technical replicates per condition.

### 4.14. Atomic Force Microscopy Analysis

AFM measurements were performed using a custom-built atomic force microscope operating in the repulsive regime of contact mode under ambient conditions and room temperature, as previously described [32]. Briefly, imaging was carried out using Bruker silicon nitride MSNL-10 cantilevers. Constant-force topographic images were acquired with an applied force of approximately 1 nN, at a typical scan rate of 2–4 s per line, and with a spatial resolution of 10 nm per pixel.

Post-acquisition data processing—including contrast enhancement, edge mask extraction, and box-counting analysis—was performed using Gwyddion 3.5 software (open-source platform for SPM data analysis) and custom scripts developed in Python (https://www.python.org/ (accessed on 11 August 2025)). 

### 4.15. AFM-Based Quantification of Cell Edge Architecture Complexity

Fractal dimension (FD) analysis offers a robust quantitative approach for assessing surface complexity in biological systems. In this study, we applied the box-counting method for FD analysis, a well-established technique for quantifying the irregularity of self-similar structures such as cell boundaries (see Appendix A, Appendix A, Appendix A).

Unlike standard Euclidean metrics, which are suited for simple geometric shapes (e.g., lines or smooth curves), FD analysis is particularly effective in capturing the complexity of biologically relevant, irregular architectures. This method has previously been used to differentiate between healthy and cancerous cervical epithelial cells based on atomic force microscopy (AFM) surface scans [60], and to evaluate boundary roughness following actin polymerization inhibition in human epithelial cells [61].

In the present work, FD analysis was applied to AFM images of U-2 OS and SAOS-2 cells. Each scan was initially processed by cropping to isolate the individual cell outline, followed by contrast enhancement to improve the accuracy of edge detection (Appendix A). A one-pixel-wide binary edge mask was then generated to represent the cell perimeter (Appendix A), enabling precise quantification of edge complexity using the box-counting approach.

Notably, high-resolution AFM imaging allowed clear visualization of lamellipodia-like protrusions approximately 100 nm in width (Appendix A), underscoring AFM’s capability to resolve sub-diffraction limit features that are typically undetectable with conventional light microscopy.

Fractal dimension values were computed from the extracted edge masks using the box-counting method, as illustrated in Appendix A. This approach enabled quantitative comparison of cell edge complexity across different treatment conditions, providing insights into cytoskeletal dynamics and morphological responses.

### 4.16. Automated Analysis of Nuclear C-Src Fluorescence

Fluorescence images were analyzed using CellProfiler (v4.2.8; Cimini Lab, Broad Institute) through an automated pipeline adapted from ImageJ-based workflows, as described [89] in The Open Lab Book [90]. Six datasets were processed, each comprising three experimental conditions (including two hFOB Ctrl-Cap conditions). For each condition, a minimum of seven paired images (nucleus and c-Src channels) were analyzed. The same pipeline was applied across all datasets, with minor adjustments made to optimize object detection when needed.

The CellProfiler pipeline files used in this study will be made available at https://cellprofiler.org/published-pipelines (accessed on 3 July 2025) upon publication. Details about the pipeline modules, including specific parameters, thresholding settings, and segmentation algorithms, are provided in the Appendix A and Methods section (Appendix A)

Images were preprocessed to correct background noise, followed by segmentation of nuclei and c-Src regions. A peripheral region was also generated around each cell to quantify background fluorescence. Nuclear morphology metrics and fluorescence intensity data were extracted from raw images.

To assess nuclear c-Src levels, Corrected Total Cell Fluorescence (CTCF) was calculated using the following formula:$$CTCF=Nuclear\;cSrc\;Integrated\;Density-\left(Background\;Mean\;Intensity\times Nuclear\;Area\right)$$
whereNuclear cSrc Integrated Density=cSrc Mean Intensity×Nuclear Area

Raw data were exported to Microsoft Excel (Microsoft 365) for CTCF calculation and further visualized and analyzed using GraphPad Prism (v10.4, San Diego, CA, USA).

### 4.17. Figure Creation

Schematic illustrations were designed using Microsoft PowerPoint, while GIMP, the GNU Image Manipulation Program, was used for image editing and post-processing. Biological components, including cell membranes, ion channels, and the nucleus, were retrieved from Bioicons [91]. Icons representing cellular features such as edge architecture, height, migration, adhesion, reorientation, and “nuclear Src” were made using the open-source software Inkscape (version 1.3.2, Inkscape Project). Other icons used were sourced from external providers: “Gears symbol” (noun-gears-7119093) by Agus Hartanto, “Chili pepper” (noun-chili-pepper-7209702) by Cherry, and “Pill” (noun-pill-6867596) by KIS, all from The Noun Project [92].

### 4.18. Data Analysis

The results are presented as the mean ± SEM. Statistical differences between means were evaluated using a parametric and unpaired Student’s *t*-test using in GraphPad Prism 9.01 software (San Diego, CA, USA). Significance levels were indicated as follows: * *p* < 0.05, ** *p* < 0.01, *** *p* < 0.001, and **** *p* < 0.0001.

## 5. Conclusions

Our study establishes TRPV1 as a central integrative hub that coordinates both mechanical and chemical signals to regulate OS cell phenotype and drug sensitivity. We demonstrate that this dual functionality is mediated by a novel TRPV1-Src-acH3 axis, which drives mechanotransduction and influences chemosensitivity in a subtype-dependent manner. Crucially, these effects are subtype-specific and absent in healthy osteoblasts, highlighting the potential for a targeted therapeutic approach with minimal off-target effects.

This discovery extends our understanding of cancer biology by highlighting a broader principle: mechanosensitive ion channels act as crucial transducers of environmental stimuli that influence not only phenotypic plasticity but also therapeutic resistance.

While in vitro studies offer promising insights, they fall short of reflecting the disease’s heterogeneity, highlighting the need for future research using advanced 3D in vitro and in vivo models. These models will enable deeper understanding of TRPV1-driven mechanotransduction within the tumor microenvironment, potentially revealing novel therapeutic strategies for aggressive, drug-resistant osteosarcomas.

## Figures and Tables

**Figure 1 ijms-26-08816-f001:**
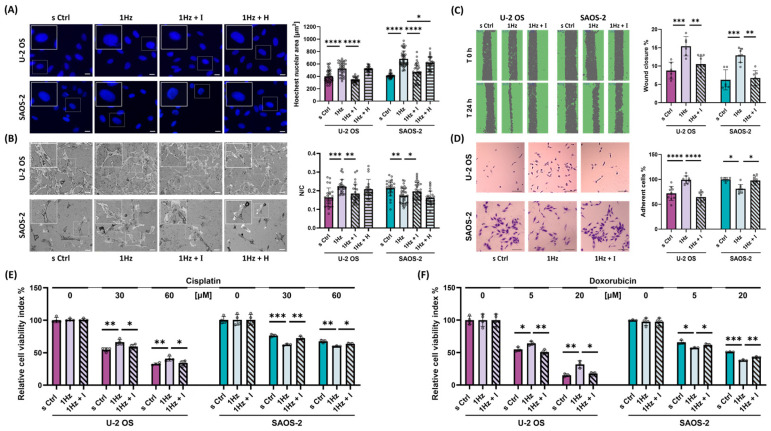
Inactivation of TRPV1 Ion Channels by AMG9810 Counteracts Mechanical Responses in OS Cells. (**A**) Representative fluorescence microscopy images of Hoechst-stained nuclei in control (s Ctrl), cyclically stretched (1 Hz), and stretched cells co-treated with AMG9810 (I) or HC030031 (H). Images were acquired at 20× magnification. Scale bar = 25 µm. Quantification of nuclear enlargement is shown as bar plots. Image analysis was performed using ImageJ 1.52. Data represent three biological replicates, with a minimum of 30 cells analyzed per condition. (**B**) Confocal microscopy images of OS cells under the same conditions mentioned above. Scale bar = 100 µm. Quantification of nuclear-to-cytoplasmic (N/C) ratio is shown as bar graphs. Data represent three biological replicates, each with at least three technical replicates and a minimum of 25 cells per condition. (**C**) AMG9810 co-treatment blocks the mechanically induced cell migration in OS cells. (**D**) AMG9810 co-treatment prevents the stretch-induced increase in adhesive capacity of U-2 OS cells. Scale bar = 100 µm. (**E**) AMG9810 alters the chemosensitivity of U-2 OS and SAOS-2 cells to cisplatin. (**F**) AMG9810 alters the chemosensitivity of U-2 OS and SAOS-2 cells to doxorubicin. Statistical analyses were performed using Student’s *t*-test. All data are derived from the best of three biological replicates, each with at least four technical replicates per condition. Statistical significance is denoted as follows: *p* < 0.05 (*), *p* < 0.01 (**), *p* < 0.001 (***), and *p* < 0.0001 (****).

**Figure 2 ijms-26-08816-f002:**
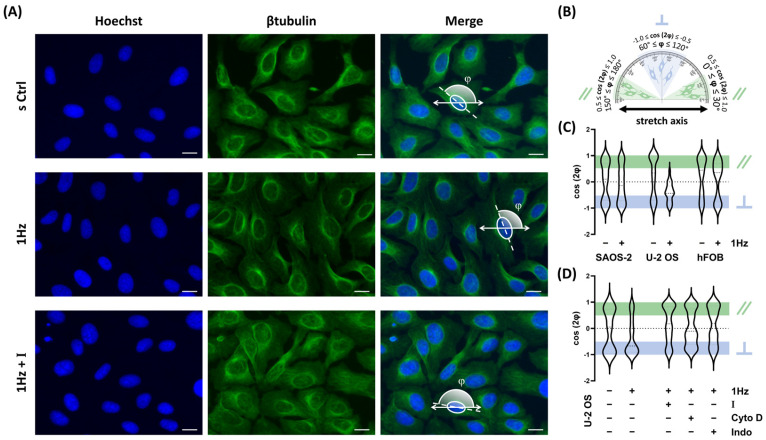
1 Hz–24 h mechanical stretch induces cell orientation in U-2 OS cells via TRPV1 activation. (**A**) Representative fluorescence microscopy images of U-2 OS cells stained with Hoechst (nuclei), β-tubulin (cytoskeleton), and merged channels. Images were acquired at 40× magnification. Scale bar = 25 μm. (**B**) Schematic representation of nuclear orientation expressed as ⟨cos 2 φ⟩. (**C**) Violin plots display the distribution of nuclear orientation values in SAOS-2, U-2 OS, and hFOB cells. (**D**) Violin plots show nuclear orientation in stretched U-2 OS cells treated with AMG (I), Cytochalasin D (Cyto D) or Indomethacin (Indo). All analyses were performed on three biological replicates, with at least 45 cells analyzed per condition.

**Figure 3 ijms-26-08816-f003:**
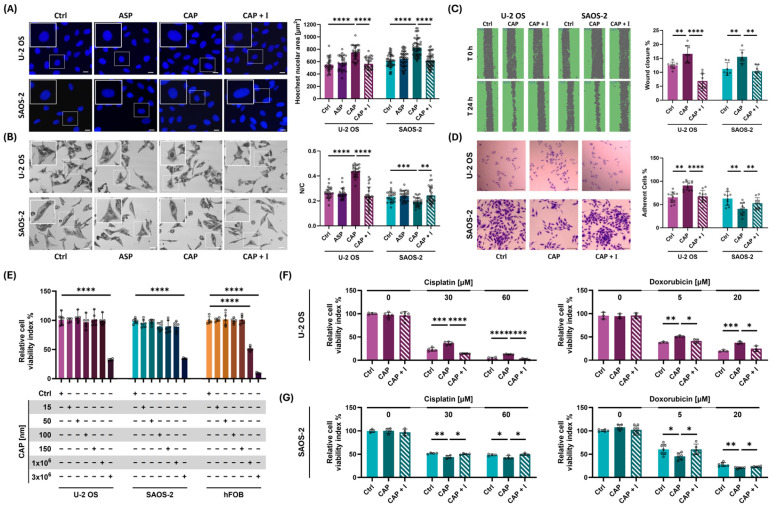
Capsaicin and its Antagonist AMG9810 Modulate OS Mechanotransduction by Activating or Blocking TRPV1 Channels. (**A**) Representative fluorescence microscopy images showing the effects of ASP7633 (ASP) and Capsaicin (CAP) with or without AMG (CAP+ I) on nuclear morphology. Magnification: 20×; scale bar: 25 µm. Nuclear enlargement quantification is shown as bar plots. Data represent three biological replicates, with a minimum of 30 cells per condition. (**B**) Representative confocal microscopy images of OS cells, with corresponding bar plot quantification of the nucleus-to-cytoplasm (N/C) ratio following the same treatments as in (**A**). Magnification: 20×, scale bar: 100 μm. The analysis includes three biological replicates with at least 25 cells per condition. (**C**) Migration assays showing the impact of capsaicin, with or without AMG9810, on the migratory ability of both OS cell lines. (**D**) The effect of Capsaicin treatment (with or without co-treatment with AMG9810) on the adhesion capability of U-2 OS and SAOS-2 cells. Scale bar: 100 μm. (**E**) Shows the relative cell viability of hFOB, U-2 OS, and SAOS-2 cell lines after treatment with increasing concentrations of capsaicin. (**F**) Relative cytotoxicity of U-2 OS cells treated with cisplatin or doxorubicin following capsaicin treatment with or without AMG. (**G**) Relative cytotoxicity of SAOS-2 cells treated with cisplatin or doxorubicin following capsaicin treatment with or without AMG. For each experiment, statistical analyses were performed on the most representative of three biological replicates, each comprising a minimum of four technical replicates per condition. Statistical significance between treated and control samples was determined using Student’s *t*-test, with significance levels indicated as *p* < 0.05 (*), *p* < 0.01 (**), *p* < 0.001 (***), and *p* < 0.0001 (****).

**Figure 4 ijms-26-08816-f004:**
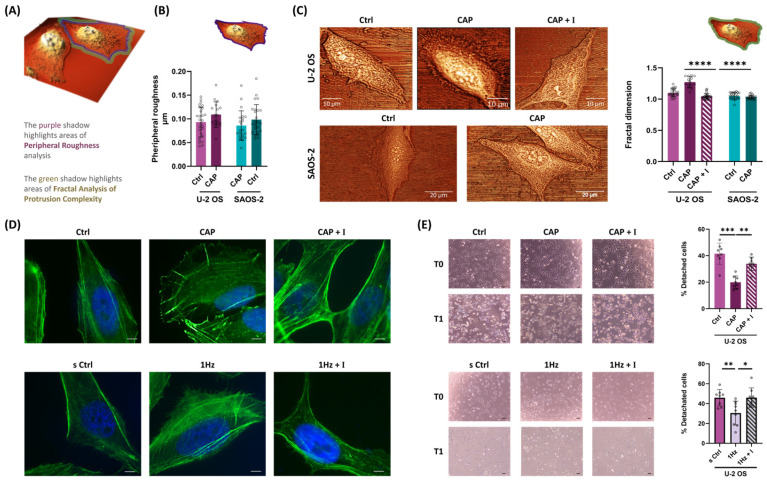
TRPV1 activation modulates OS cell peripheral ultrastructure, altering cell adhesive properties. (**A**) Image represents topographic AFM scans of OS cell acquired on rigid glass substrates. (**B**) Quantification of peripheral roughness in U-2 OS and SAOS-2 cells treated with capsaicin on rigid glass substrates. (**C**) Fractal dimension (FD) analysis of cell edge complexity, based on high-resolution AFM images of OS cells, treated with capsaicin with or without AMG9810. (**D**) Immunofluorescence staining of filamentous actin (phalloidin) in U-2 OS cells subjected to mechanical or chemical activation of TRPV1, with or without co-treatment with AMG9810. Images at 100×. Scale bar = 25 µm. (**E**) Quantification of cell detachment after the same treatment as in (**D**). All experiments were performed in triplicate, with at least eight biological replicates per condition. Statistical significance was determined using Student’s *t*-test. Significance levels are indicated as follows: *p* < 0.05 (*), *p* < 0.01 (**), *p* < 0.001 (***), and *p* < 0.0001 (****).

**Figure 5 ijms-26-08816-f005:**
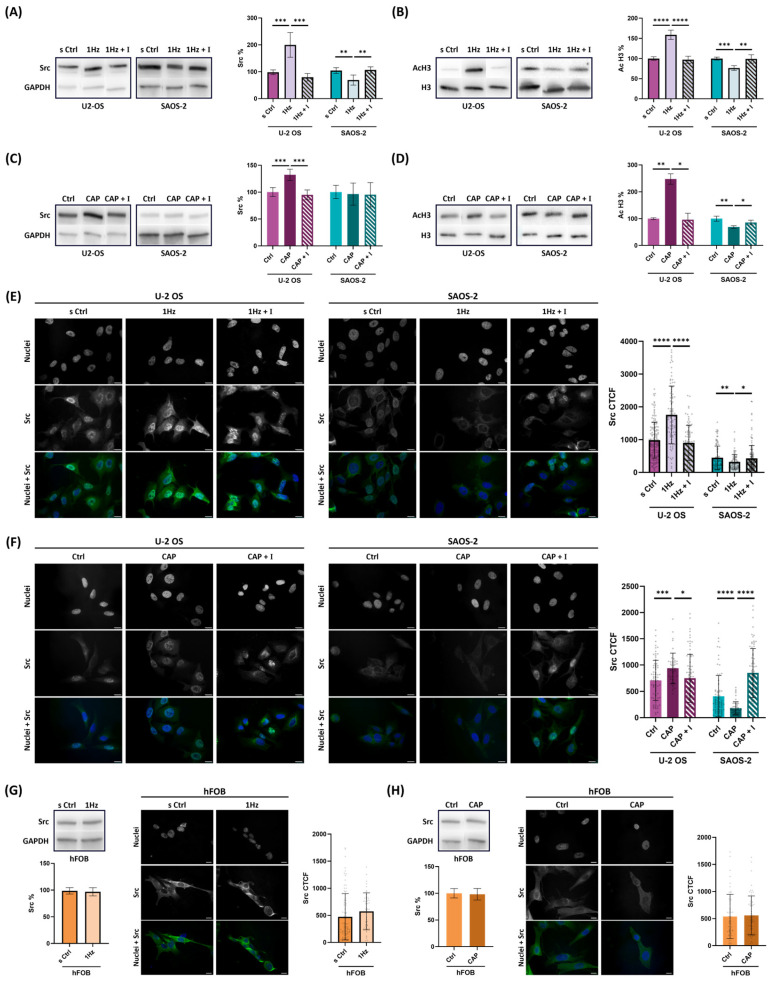
TRPV1 activation modulates expression and nuclear localization Src in OS cells. (**A**–**D**) Representative Western blot images and their respective densitometric analyses of cell extract from U-2 OS or SAOS-2. (**A**,**C**) SRC (60 kDa) and GAPDH (37 kDa) protein levels following mechanical stretch or capsaicin treatment, with or without co-treatment with AMG9810, respectively. (**B**,**D**) Acetylated histone 3 (acH3, 17 kDa) and Histone 3 (H3 17 kDa) levels under the same conditions as above. (**E**,**F**) Representative immunofluorescence images of nuclei (Hoechst), Src protein, and merged channels in OS cells subjected to mechanical stretch or capsaicin treatment, with or without co-treatment with AMG9810. Src nuclear levels quantified by corrected total cell fluorescence (CTCF) are reported as bar plots (mean ± SD). (**G**,**H**) Western blot and CTCF analysis of Src in hFOB cells under mechanical stretch or capsaicin treatment. Densitometric analysis of the Western blot bands was performed using ImageJ 1.52 software and quantified in arbitrary units. CellProfiler 4.2.8 was used to identify nuclear SRC fluorescence intensity and area values (Appendix A). Data represent three biological replicates, with a minimum of 45 cells analyzed per condition. Error bars represent the standard error of the mean (SEM). Statistical analysis was conducted on at least three biological replicates. Statistical significance was determined using an unpaired *t*-test, with results shown as mean ± SD; *p* < 0.05 (*), *p* < 0.01 (**), *p* < 0.001 (***), and *p* < 0.0001 (****).

**Figure 6 ijms-26-08816-f006:**
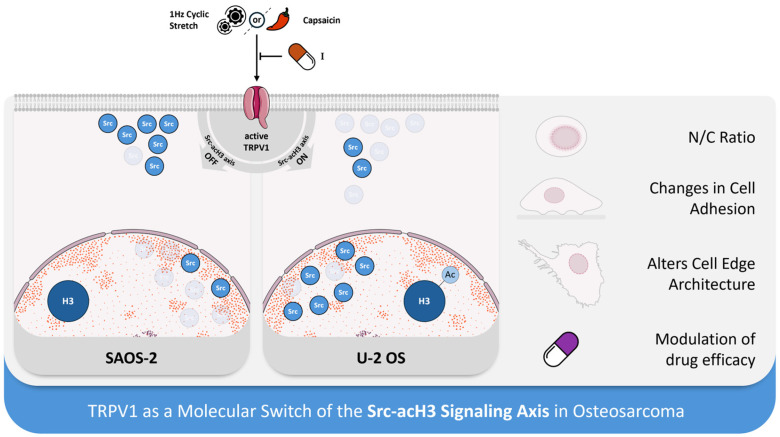
Graphical representation of the proposed mechanism by which TRPV1 regulates the Src–acH3 mechanosignaling axis in osteosarcoma cells. TRPV1 appears to play a central role in modulating histone3 acetylation by influencing the subcellular localization of Src, thereby contributing to the phenotypic divergence observed between osteosarcoma subtypes. In the SAOS-2 cell model (left), TRPV1 activation downregulates the Src–acH3 axis, resulting in reduced nuclear Src levels. In contrast, in the U-2 OS model (right), TRPV1 activation upregulates the axis by promoting nuclear translocation of Src. ‘I’ denotes AMG, the competitive inhibitor of TRPV1.

## Data Availability

All data generated or analyzed during this study are included in this published article and its Appendix A.

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
