# Peer review of "A ‘Spicy’ Mechanotransduction Switch: Capsaicin-Activated TRPV1 Receptor Modulates Osteosarcoma Cell Behavior and Drug Sensitivity"

_ijms, 2025, doi:10.3390/ijms26188816_

Round 1
Reviewer 1 Report
Comments and Suggestions for Authors
On request of IJMS, I have revised the manuscript titled “A 'Spicy' Mechanotransduction Switch: Capsaicin-Activated TRPV1 receptor modulates Osteosarcoma Cell Behavior and Drug Sensitivity “, by Arianna Buglione et al.
Aiming at in deep exploring the role of mechanical stimuli in OS progression, still to little investigated so far, here, the Authors examined the transient receptor potential vanilloid 1 (TRPV1) channel functions, finding that it acts as a key regulator of mechanotransduction and drug responsiveness in OS cells. First, they discovered that differently from less aggressive SAOS-2 tumour cells, the aggressive ones (U-2 OS) undergo TRPV1-dependent perpendicular reorientation and that the TRPV1 agonist capsaicin alters metastatic traits of OS cells, such as cell adhesion, edge architecture, migration rate, nuclear-to-cytoplasmic ratio, and chemotherapeutic sensitivity to doxorubicin and cisplatin, in a different way. TRPV1 activation by capsaicin elicits subtype-specific effects absent in healthy hFOB osteoblasts. These TRPV1-mediated effects were not observed in non-malignant cells, underscoring their tumour specificity activity.
General Comments
Considering that osteosarcoma (OS) is the most common primary malignant bone tumour arising in the highly mechanosensitive bone tissue, characterized by marked heterogeneity and resistance to conventional therapies, the findings of this study appear interesting and relevant. Mechanobiology is an emerging science, which could have great potential in controlling the development and progression of cancers, including cell proliferation, invasion, and therapy resistance. The standardized uniaxial cyclic stretch protocol previously developed by Authors and used here is an excellent tool to in vitro study how bone-derived tumours respond and reorient themselves to cyclic mechanical loading. On a scenario where, while molecular drivers of OS have been extensively characterized, but the role of mechanical stimuli in OS progression remains underexplored, the Authors, applying their protocol, have discovered that TRPV1 receptor activated by spicy capsaicin functions as a mechanosensitive signalling hub that integrates mechanical and chemical cues to drive epigenetic remodelling and phenotypic plasticity in OS, offering a potential therapeutic target—specifically in aggressive, drug-resistant subtypes. It was a pleasure for me to read this work, which appeared well written, easy to read and understand, and loaded with robust results and relevant findings.
I suggest IJMS the publication of this study after having addressed some my curiosities and some very minor issues.
According to previous studies, U-2 OS cells were negative for most of the investigated osteoblastic markers and demonstrated no bone alkaline phosphatase activity (ALP), differently from SAOS-2 (PMID: 15736406), which are characterized by a high bone ALP action (https://doi.org/10.1002/jbmr.5650020310). Could these differences in the two cell lines influence their responses to mechanical stimuli and capsaicin-activated TRPV1-mediated effects? Is there some essay to assess this?
Can the better understanding of the biomechanical aspects of OS be useful in ameliorating OS early diagnosis or in the development of more sensitive diagnostic tools?
Minor
Please, remove capital letters from keywords.
Figure 1-5. The captions are very long. Please, move part of their content to the main text, when it is not a text duplication. Conversely, just remove it.
Dear Authors,
I am aware that my comments should make me ask for minor revisions, but, since I am interested in receiving responses to my curiosities, I asked for major revisions, in order to have the possibility to reconsider this study and your valuable reply. I apologies for this.
Author Response
Response to Reviewer 1
We sincerely thank Reviewer 1 for the thoughtful and encouraging evaluation of our manuscript entitled “A 'Spicy' Mechanotransduction Switch: Capsaicin-Activated TRPV1 receptor modulates Osteosarcoma Cell Behavior and Drug Sensitivity”. We are pleased that the reviewer found our study interesting, well written, and scientifically robust. We also appreciate the insightful comments and questions, which allowed us to further refine the manuscript and clarify several aspects. Below, we provide a point-by-point response to each of the reviewer’s comments and suggestions.
Major Comments / Questions
Comment 1 :
According to previous studies, U-2 OS cells were negative for most of the investigated osteoblastic markers and demonstrated no bone alkaline phosphatase activity (ALP), differently from SAOS-2 (PMID: 15736406), which are characterized by a high bone ALP action (https://doi.org/10.1002/jbmr.5650020310). Could these differences in the two cell lines influence their responses to mechanical stimuli and capsaicin-activated TRPV1-mediated effects? Is there some assay to assess this?
Response to comment 1:
We thank the reviewer for highlighting this important distinction. Indeed, U-2 OS and SAOS-2 cells exhibit notable differences in their osteoblastic differentiation status, which likely influence both their mechanosensitivity and TRPV1-mediated responses.
As noted, SAOS-2 cells are characterized by high alkaline phosphatase (ALP) activity and robust expression of osteoblastic markers, representing a more mature osteoblast-like phenotype. Our previous work (Alloisio et al., 2023) demonstrated that these cells did not exhibit a significant response to 24-hour cyclic mechanical stimulation at 1 Hz, as indicated by stable gene expression and ALP activity levels. In contrast, U-2 OS cells, which are less differentiated and exhibit minimal ALP activity, reflect a more aggressive, undifferentiated tumor phenotype. Consistent with this, our preliminary data (unpublished) show that U-2 OS cells also failed to respond to the same cyclic stimulation protocol, with no detectable changes in real-time qPCR or ALP activity assays.
These intrinsic differences between the two cell lines could explain the distinct cellular responses observed in our study. It is plausible that U-2 OS cells, owing to their less differentiated and more aggressive phenotype, have altered cytoskeletal organization, integrin expression, and mechanotransduction pathways, which could impact TRPV1 activation in response to mechanical or chemical stimuli. Such mechanisms may account for the subtype-specific effects of TRPV1 observed in our experiments.
To further investigate the role of differentiation status in TRPV1-mediated mechanotransduction, future studies could incorporate assays such as ALP activity measurements and profiling of osteogenic markers like RUNX2, OCN, and OPN, in osteogenically-induced cells. Previous studies have shown that these markers, including ALP, OCN, and COL, correlate with cytoskeletal organization and cellular mechanotransduction (Zhao Y, Sun Q, Huo B. Focal adhesion regulates osteogenic differentiation of mesenchymal stem cells and osteoblasts. Biomater Transl. 2021 Dec 28;2(4):312-322).
Although this topic is beyond the scope of the current manuscript, we agree that it offers a promising avenue for future research. While we have not modified the manuscript to address these points in detail, we appreciate the reviewer’s suggestion and will consider this direction in subsequent studies.
Ccomment 2:
Can the better understanding of the biomechanical aspects of OS be useful in ameliorating OS early diagnosis or in the development of more sensitive diagnostic tools?
Response to comment 2:
Thank you for this insightful comment. While our study primarily focuses on mechanotransduction and drug responsiveness, we agree that the broader implications of OS mechanobiology may extend to diagnostics.
Understanding how OS cells sense and respond to mechanical stimuli—both in vitro and in vivo—through TRPV1 and other channels could help identify early phenotypic markers of aggressive behavior, such as stretch-induced reorientation, altered adhesion dynamics, or cytoskeletal remodeling. These features may be leveraged in biophysical screening platforms (e.g., microfluidic devices or mechanosensitive biosensors) for early detection or stratification of tumor aggressiveness.
Although current data are limited, observed differences in TRPV1 expression or activity across OS cell lines and in vivo models may reflect heterogeneity in mechanotransductive signaling. If validated, this variability could inform the development of more refined diagnostic or prognostic tools based on biomechanical profiling.
We have now addressed this point in the revised Discussion section ( line 633 and line 710), highlighting the potential of biomechanical profiling to complement molecular diagnostics in OS.
Minor Comments
Comment 3:
Please, remove capital letters from keywords.
Response to comment 3:
We have corrected the formatting by removing capital letters from the keywords section, as requested.
Comment 4:
Figures 1–5: The captions are very long. Please, move part of their content to the main text, when it is not a text duplication. Conversely, just remove it.
Response to comment 4:
Thank you for this helpful suggestion. We have revised all figure captions (Figures 1–5) to make them more concise. Descriptive content that provided contextual or interpretative information has been moved to the main text, while redundant or unnecessary text has been removed from the captions.
Final Note
We are pleased that our manuscript has sparked Reviewer 1's curiosity. We greatly appreciate the reviewer’s kind words and recognition of our study's novelty and potential impact. We hope that our revisions and clarifications adequately address all concerns and curiosities. We are truly grateful for your time and insightful comments, which have significantly contributed to improving the quality of our manuscript.
Sincerely,
Prof. Magda Gioia
(on behalf of all co-authors)
Reviewer 2 Report
Comments and Suggestions for Authors
This manuscript investigates the role of the TRPV1 mechanosensitive ion channel in osteosarcoma (OS) cell mechanotransduction, focusing on its regulation of cell morphology, adhesion, migration, and chemotherapeutic sensitivity. The study integrates confocal microscopy, atomic force microscopy, biochemical assays, and functional analyses, providing a comprehensive exploration of how TRPV1 integrates mechanical and chemical cues to influence osteosarcoma cell behavior.
This work is surely within the scope of IJMS, was submitted to the proper Section, with the vey low value of iThenticate scor indicating high level of originality in manuscript preparation. This work is of high overall quality. However, I have also some questions and comments for the authors, presented below.
Major comments:
Abstract is slightly too long, it should be shortened.
The same applies to introduction, while it is interesting it is simply too long.
Line 151, When introducing capsaicin as the TRPV1 agonist, it would strengthen the rationale to briefly mention why TRPA1 agonists were included for comparison, to better frame the study’s design.
Line 214, the cisplatin/doxorubicin assays are presented clearly, but it would be beneficial to report exact n values for technical replicates and indicate whether the experiments were blinded to reduce bias.
Line 325, The fact that capsaicin affects OS cells but not hFOBs is important—consider adding quantitative statistical data for hFOBs in the main text
Line 681, The discussion on morphological remodeling differences between cell lines is valuable; however, please consider presenting possible molecular determinants (e.g., differential expression of cytoskeletal regulators) to give more depth analysis.
Lines 970-973, while the authors have stated which software has been used (GraphPad), they also are required to precisely state what kind of statistical tests have been used for comparison. Tukey?
Line 702, at this point a separate and CONCISE conclusions section should be created, the part of the discussion can be moved to conclusions. Also, please shortly describe the limitations of the current study.
Minor points:
Line 18, there’s a typo, it should be “identify”
Line 722, it should be “CO2”
Author Response
Response to Reviewer 2
We would like to express our sincere thanks to Reviewer 2 for the careful consideration of our manuscript, and for the positive feedback highlighting the originality and comprehensive nature of our approach. We greatly appreciate the recognition of the overall quality and innovation presented in the work.
Below, we address each of your comments in detail, outlining the corresponding revisions made to the manuscript. We trust that these changes enhance the clarity and depth of the paper.
Major Comments
- Abstract is slightly too long, it should be shortened.
Response:
We agree with the reviewer. The abstract has been shortened to improve clarity and conciseness while retaining all key findings and methodologies. - The same applies to the introduction, while it is interesting it is simply too long.
Response:
Thank you for this suggestion. We have revised and condensed the Introduction section to maintain focus on the most relevant background information, reducing its length without compromising necessary context. - Line 151 – Capsaicin as TRPV1 agonist: rationale for inclusion of TRPA1 agonists.
Response:
We appreciate the referee’s comment and the opportunity to clarify our rationale. We have revised the text accordingly in the updated manuscript (at line139).
Our previous research demonstrated that LE135, a compound that activates both TRPV1 and TRPA1, can effectively mimic the effects of mechanical stimulation in human osteosarcoma (OS) cells. Based on these findings, our rationale for including agonists targeting both channels was to further explore the potential for chemically modulating mechanosensitive (MS) signaling in OS.
Specifically, we aimed to determine whether the observed effects of LE135 were attributable to activation of TRPV1, TRPA1, or a combined effect of both. Since TRPV1 (the target of capsaicin) and TRPA1 are both implicated in MS signaling pathways involved in cancer, it was essential to include selective agonists for each channel. This approach allowed us to systematically dissect their individual and synergistic contributions to mechanotransduction in OS cells
- Line 214 – Cisplatin/doxorubicin assays: report n values and whether experiments were blinded.
Response:
The statistical analyses for the cytotoxicity assays shown in the figure were performed using the best of three technical replicates, each comprising four biological replicates, as indicated in the figure caption. In response to the referee's suggestion, the exact numbers of technical and biological replicates for each drug assay have been added to Section 4.5 of the Materials and Methods for enhanced clarity. - Line 325 – Capsaicin effects in OS cells vs. hFOBs: add statistical data for hFOBs.
Response:
In response to Referee 2’s suggestions, quantitative statistical data for hFOBs have been added to the main text (at line 317) and corresponding figure legends to support the statement regarding the differential response between OS cells and hFOBs. - Line 681 – Morphological remodeling: suggest possible molecular determinants.
Response:
We thank the reviewer for this excellent suggestion. We have expanded the discussion at line 680 to propose potential molecular determinants, including differential expression of cytoskeletal regulators such as Rho GTPases and actin-binding proteins, which may account for the observed differences in remodeling between the cell lines. - Lines 970–973 – Specify statistical tests used.
Response:
We thank the reviewer for this comment. As noted in the revised manuscript (Section 4.18 of the Materials and Methods), statistical differences between two groups were assessed using the parametric unpaired Student’s t-test. For experiments involving more than two groups, we performed pairwise comparisons using the same test.
We acknowledge that performing multiple t-tests without correction can increase the risk of Type I error. However, given the limited number of comparisons in our experiments and the exploratory nature of the study, we opted to prioritize simplicity and interpretability of results. We have now explicitly clarified this in the Methods section.
- Line 702 – Add a concise Conclusions section and state study limitations.
Response:
A separate and concise Conclusions section has been revised in accordance with Referee 2’s suggestions, by revising the Discussion at line 705. Additionally, we have included a brief discussion of the study’s limitations, including the need for in vivo and 3D in vitro validation, as well as the limited range of OS subtypes analyzed (at line 989).
Minor Points
- Line 18 – Typo “indentify” corrected to “identify”.
Response:
Corrected as suggested. - Line 722 – “CO2” corrected.
Response:
Thank you for catching that. The typo has been corrected to “CO₂”
Once again, we sincerely thank the reviewer for their valuable comments, which have significantly improved the clarity and quality of our manuscript.
Sincerely,
Prof. Magda Gioia
(on behalf of all co-authors)
Round 2
Reviewer 1 Report
Comments and Suggestions for Authors
Dear Authors,
thank you very much for your explanations and revision. While renewing my congratulations on your study, it is my pleasure to inform you that your paper has been accepted for publication.
Reviewer 2 Report
Comments and Suggestions for Authors
The authors have revised and improved their work - current version can be accepted.